# Extracellular Vesicles of Mesenchymal Stem Cells: Therapeutic Properties Discovered with Extraordinary Success

**DOI:** 10.3390/biomedicines9060667

**Published:** 2021-06-10

**Authors:** Gabriella Racchetti, Jacopo Meldolesi

**Affiliations:** 1Division of Neuroscience, San Raffaele Institute, Via Olgettina 58, 20132 Milan, Italy; racchetti.gabriella@hsr.it; 2Department of Neuroscience, Faculty of Medicine, Vita-Salute San Raffaele University, Via Olgettina 58, 20132 Milan, Italy

**Keywords:** clinical developments, clinical trials, engineering, ectosome and exosome, heterogeneity, immune reactions, inflammation, luminal cargo, miRNA, miR-, protection of organs, senescence, regenerative medicine, stromas employed: from bone marrow, adipose tissue, umbilical cord, blood

## Abstract

Mesenchymal stem cells (MSCs), the cells distributed in the stromas of the body, are known for various properties including replication, the potential of various differentiations, the immune-related processes including inflammation. About two decades ago, these cells were shown to play relevant roles in the therapy of numerous diseases, dependent on their immune regulation and their release of cytokines and growth factors, with ensuing activation of favorable enzymes and processes. Such discovery induced great increase of their investigation. Soon thereafter, however, it became clear that therapeutic actions of MSCs are risky, accompanied by serious drawbacks and defects. MSC therapy has been therefore reduced to a few diseases, replaced for the others by their extracellular vesicles, the MSC-EVs. The latter vesicles recapitulate most therapeutic actions of MSCs, with equal or even better efficacies and without the serious drawbacks of the parent cells. In addition, MSC-EVs are characterized by many advantages, among which are their heterogeneities dependent on the stromas of origin, the alleviation of cell aging, the regulation of immune responses and inflammation. Here we illustrate the MSC-EV therapeutic effects, largely mediated by specific miRNAs, covering various diseases and pathological processes occurring in the bones, heart and vessels, kidney, and brain. MSC-EVs operate also on the development of cancers and on COVID-19, where they alleviate the organ lesions induced by the virus. Therapy by MSC-EVs can be improved by combination of their innate potential to engineering processes inducing precise targeting and transfer of drugs. The unique properties of MSC-EVs explain their intense studies, carried out with extraordinary success. Although not yet developed to clinical practice, the perspectives for proximal future are encouraging.

## 1. Introduction

All types of cells are known to express a peculiar form of secretion consisting in the release of two types of vesicles, the small exosomes (diameter between 50 and 150 nm) and the larger ectosomes (called also microvesicles, diameter between 100 and 400 nm). Exosomes are generated and accumulated within specialized endosomal vacuoles, the multi-vesicular bodies (MVB). The release of exosomes occurs upon exocytosis of the latter vacuoles Ectosomes are assembled at plasma membrane rafts, which undergo pinching off and then shedding for secretion [1,2,3,4] (Figure 1). Successful procedures for separation of the two, small and large vacuoles have become widely available at the beginning of our century. Therefore, exosomes and ectosomes can be investigated separately. For this they are often named small and large extracellular vesicles (sEVs and lEVs) [5,6]. In other cases, the studies are being made on the two types of vesicles isolated together, which are usually named extracellular vesicles, EVs. The present review deals with the mixed vesicles specified according to the nomenclature [3,4,5,6,7].

Both exosomes and ectosomes are composed by two structures: a membrane delimiting their lumen, with proteins often exposed at both its external and internal surfaces; and the luminal cargoes, composed by various proteins, lipids, coding and non-coding RNAs, accompanied, in some cases, by DNA sequences [1,2,3]. In term of composition, exosomes and ectosomes include, together with some specific molecules, also many common membrane and cargo molecules often present also in vesicles secreted by different parental cells. The largely common molecules may explain the general similarity of the two types of vesicle from various cell origin, while the specific molecules may account for their differences. The latter may include molecules that govern at least some EV specificities, such as the direct binding to their target cells. The dependence of some specific EV properties on their parental cells has been demonstrated in many cases. For example, in the brain a few specific properties have been found to differentially characterize the EVs secreted by neurons, astrocytes and microglia [8].

Upon their secretion and some navigation in the extracellular space, EVs often establish specific autocrine or paracrine interactions, the latter to their target cells, mediated by the binding to receptor/surface or endosomal proteins, followed by fusion. At various steps of their pathways, EVs contribute to the molecular specificity of their interacting cells: of parental cells by their secretion; of target cells by their fusion and ensuing discharge of cargo components [1,2]. Alternatively to fusion with target cells, EVs can move to external fluids, such as the cerebrospinal fluid and the blood serum. Such fluid targeting is important for the identification and analysis of specific biomarker molecules, and also for various EV functions [9]. Since the role of EVs in external fluids is not critical in this review, it is not going to be mentioned any more.

The functional activity of many EVs has been compared to the overall activity of their parental cells. Often the EV activity, although relevant, is only partial, in the sense that many functions of parental cells do not appear in their EVs. Such conclusion is valid for many, but not for all types of cells. The family of mesenchymal stem cells (MSCs), generated and resident in the stroma of all organs, have been shown to secrete EVs, the MSC-EVs, playing key roles in most of their functions. Interestingly, among the common functions of MSCs and MSC-EVs is the well known therapy of many diseases, relevant not only for the pathology but also for future medical developments. This review is focused on MSC-EVs, illustrated for many of their functions, in particular for those recognized during the last few years, including their therapy for 7many diseases.

## 2. Mesenchymal Stem Cells and Their Extracellular Vesicles

MSCs, the cells of the stroma in all body tissues, had been discovered a few decades ago. Since then they have been isolated and progressively investigated. Their functions include self-replication and multidirectional differentiation, participation in physiology and in pathology of diseases, with manipulation of immunological processes and attenuation of inflammatory processes. Moreover, since over two decades MSCs are known to play a role in the therapy of many diseases, envisaged also for possible developments in clinical medicine [10,11]. Initially, MSCs where shown to play the role of recombinant proteins in bone and cartilage healing [12]. Shortly thereafter they were recognized as progenitors of osteoblasts, chondroblasts and fibroblasts, a series of cells that “hold us together” [13]. Somewhat later, analogous results were obtained, concerning however local processes such as self renewal and repair, followed by therapeutic effects on several organ diseases: of lung, heart, kidney, brain, cancers, and others [10,11,14,15,16]. Such therapeutic effects were thought started by paracrine fusions of MSCs with their target cells. Such fusions, however, were not dependent on MSC alone. They are reinforced by soluble and bioactive factors, such as cytokines and growth factors, released by the cells in parallel to their fusions [17,18]. The identification, around 2010, of the MSC-EV secretion suggested their possible participation in the MSC effects, a cooperation confirmed by experiments in which cells and vesicles were administered together [19]. In most initial experiments, carried out in animals, MSCs were shown to induce positive results in various types of cell therapy. The extension to pathology and therapy of human diseases was therefore envisaged and intensely investigated. Yet, despite the encouraging preclinical outcomes in animal models, the risk of human MSC infusion and transplantation therapies turned out to be considerable, including immune rejections and various types of cancer initiation and promotion [20]. Because of the negative results, the enthusiasm of MSC decreased rapidly, with ensuing reduction of its employment in therapy. In parallel, the majority of registered clinical trials applying MSC therapy to diverse human diseases did fall short of expectations. At present, therefore, the study of these therapies remains of interest, however with clinical perspectives limited to only a few types of diseases [21,22,23].

The problems recognized to MSC employment, especially in human therapies, has opened the way to corresponding studies about vesicles. MSC-EVs recapitulated many of the therapeutic effects of MSCs, with improvements of considerable importance. Emerging evidence has shown that, in many kinds of diseases, MSC-EV treatments have equal or even better efficacies than MSCs, with a considerable decrease of risks. For many therapies, therefore, EVs replaced their parental cells. Such demonstrations have opened the chance to a new type of therapy, widely known as cell-free therapy, based on the use of MSC-EVs [24]. The advantages of such development are considerable, including increased safety and faster tissue penetration. In addition, the inability of MSC-EVs to self-replicate greatly reduces the risk of tumors and expansions, typical of MSCs; the limited potential to trigger the immune system prevents disappointments even in the course of allo- and xeno-grafts; and easier transport and storage makes the potential of EV therapy optimal when compared to standard cell-based approaches (Figure 1) [19,20,22,24,25,26]. These results have contributed to the development of some clinical studies. Applications of MSC-derived EVs have gained increasing interest, as many risks of the MSC-based therapy are avoided. Clinical studies, however, are still early, meaning that the work is encouraged but not yet carried out in detail. The discovered MSC-EV properties, confirmed in a variety of experimental conditions, have lead to an explosion of interest about these vesicles. At present the number of published articles is over 1000, grown from a few tens in 2015 to over 350 in 2020.

A critical property of MSC-EVs is heterogeneity. As already mentioned, their parental cells are generated in the stroma of tissues, which may be variable. However, the EVs employed for investigation do not come from all stromas but concentrate on only four of them, chosen based on their accessibility and personal preference: bone marrow, adipose tissue, umbilical cord and blood from that cord. Variability reported among these EVs is significant, concerning components such as proteins and, especially, non-coding RNAs that participate in the generation of properties such as potency, inflammatory resolution and tissue regeneration [27,28,29]. Diseases differentially affected by various MSC-EVs are numerous. For example the vesicles of the umbilical cord affect more acute diseases and operate well in damage repair; while those from adipose tissue are more active against prolonged diseases and immune responses, including Alzheimer’s disease and multiple sclerosis; those from bone marrow are especially active in tissue regeneration [28,30]. The relation with diseases, useful also for preclinical analyses, stimulated the use of MSC-EVs as therapeutic tools of translational potential [30,31,32].

An unexpected but critical role of MSC-EVs concerns cell aging. Senescence is characterized by loss of proliferative potential, resistance to cell death by apoptosis, and expression of a secretory phenotype including pro-inflammatory cytokines and chemokines, tissue-damaging proteases and growth factors, all contributing to tissue alteration and loss of homeostasis [33]. Treatment with MSC-EVs reduces senescence, in culture and in vivo, and improves health span, an effective and safe approach conferring effects of adult stem cells, avoiding the risks of tumor development and donor cell rejection [34]. The mechanism of such protection includes down-regulation of superoxide dismutases with elevation of reactive oxygen species [35]. Cell rejuvenation has been confirmed in various cell types including endothelial and muscle cells of arteries with decreased hypertension [36,37], bone marrows [38], and others.

## 3. MSC-EVs Express Also Functions Shared by EVs from Other Parental Cells

Although peculiar in many respects, MSC-EVs are EVs. Therefore, their properties shared by the other types of EVs, in particular those dependent on paracrine fusions with target cells, cannot be a surprise. Such fusions include the transfer of molecules concentrated in their cargos, including several bioactive molecules, transcription factors and enzymes. Additional transferred molecules of importance are the non-coding miRNAs, essential for many therapies (Table 1), which will be mentioned in following Sections. The cell free transfer can induce comprehensive effects including the in vivo expansion of target cells, such as the hematopoietic stem cells [39]. These expansions and other effects contribute to the development of translational medicine [40].

Transfer, however, can contribute also to the distribution and function of organelles, for example mitochondria. The mechanism involved can be various. Some evidence supports the direct transfer of the organelles, accumulated together with molecules during the assembly of vesicles, ultimately discharged into the cytoplasm of target cells. Such an effect, conditioned by the MSC environment, suppresses other functions of EVs such as oxidative stress and apoptosis. In other cells the effect of MSC-EVs on mitochondria is indirect, dependent on a reinforcement of the mitochondrial transcription factor TFAM, with ensuing stabilization of mitochondrial DNA, attenuation of the mitochondrial damage and of inflammation [41,42,43]. A consequence of these events is an improvement of the barrier integrity with protection of cells from excess cytokines, which in turn can induce a severe in vivo danger, the acute respiratory distress syndrome (ARDS l [44]).

## 4. Immunity

To start our presentation about processes of physiology and pathology we focused on immunity, a preliminary field of great interest. Experience with MSC-EVs has demonstrated that the use of these vesicles is much safer and more effective than that of MSCs themselves. Among their functions, MSC-EVs modulate immune responses. Recruitment in their proximity of needed cells may result in boosting immune responses, associated in many cases to protective roles in infectious diseases. These actions have offered appreciable advantages [45] in the differentiation, activation, and proliferation processes of immune cells [25,46,47]. Immune-modulatory and regenerative functions are induced in the course of both the innate and adaptive immune reactions. The cells involved include T and B lymphocytes, natural killers, dendritic cells, and macrophages, modifying their polarization. The effects induced by MSC-EVs, including anti-inflammatory, anti-aging and wound healing, play critical roles in a variety of diseases [47,48,49].

As far as the mechanisms, MSC-EV immunomodulation is governed by proteins, including growth factors, interleukins, and also miRNAs, such as miR-21-5p, miR-223, miR-146a and miR-199a, whose expression is increased [45,49,50]. In a mouse model, treatment with MSC-EVs ameliorated significantly the immune destructions and confirmed the importance of the non-coding miRNAs also in the clinical developments [50]. Other treatments exacerbate various aspects of neuropathology. The therapeutic potential of MSC-EVs towards the alleviation of pathological (trained) immunity is mediated by the factors already mentioned. Also in the brain, accessed immunity is often mitigated by MSC-EV-based regenerative actions [51]. The use of MSC-EVs for therapy of diseases has become relevant in many cases including encephalomyelitis and multiple sclerosis of the brain. The same vesicles also provide more potent therapeutic strategies for other immune-related disorders [50,52,53].

## 5. Diseases: Mechanisms of Protection by MSC-EV

Many of the immunological events mentioned in the previous Section have role in diseases and in their protection by MSC-EVs. It is clear, therefore, that MSC-EVs modulate a large number of signaling pathways in many tissues. For example, they prevent delayed injury, enhance parenchymal remodeling and make tissue recovery possible [54,55]. Ideally, such effects should be based on the mechanisms of action of the vesicles. This, however, is a challenge because the efficacy of the vesicles is considerable and occurs against multiple diseases with various types of pathology. We conclude that the action of MSC-EVs is complex, different for each disease and difficult to elucidate [55].

At this point we will start presenting the diseases affected by MSC-EVs, relevant by their cell-free action. The diseases are quite different from each other, also in terms of their active drugs. The therapeutic effects of the various EVs greatly depend on their cargo components. The commonalities of their MSC-EV therapy can be explained by their specific drugs, strengthened by the vesicles via the activation or the depression of various biomedical protein factors and, even more, of specific miRNAs (Table 1). In this Section we will illustrate the state of numerous diseases in which the MSC-EV therapy has been demonstrated [56,57,58].

The diseases of several organs and processes: bones, heart and vessels, kidneys, brain, cancers and COVID-19, are discussed here in some detail. Additional diseases, such as cutaneous wounds, digestive diseases, acute/mature lung injuries, diabetes, depression and others, also affected by MSC-EVs, have not been included here. The same has been decided for cancers, intensively investigated inducing excellent results, that will be reported elsewhere. The data missing here can be found in other reviews of the literature [47,56,57].

### 5.1. Bones

Bones are the organs for which regulatory control by MSC had been among the first reported at the beginning of 2000 [15]. Progress of knowledge has emphasized the role of MSC-EVs. Osteoblastogenesis stimulation depends on miRNAs such as 133-3p for EVs from bone marrow stem cells (Table 1), 125b-2-3p for those from adipose stem cells [59]. MSC-EVs have been identified for their cell-free therapy of rheumatic diseases, including arthritis and osteoarthritis, dependent on their low immunogenicity and control of inflammatory factors (miR-483-5p, Table 1). The enormous potential of EVs for treatment of rheumatism is still to be investigated for clinical application [60]. Finally local bone diseases, including inflammation, influence bone regeneration (miR-378, Table 1). In this case the MSC-EVs protection is mediated by their interaction with macrophages regulated by miRNAs active in positive and negative immunity regulation [61].

### 5.2. Heart and Vessels

Cardiovascular diseases are a leading cause of morbidity and mortality, accounting for approximately one third of deaths every year, caused mostly by myocardial infarction induced by ischemia followed by reperfusion. Studies of the last few years demonstrated that two distinct miRNAs, miR-125b and miR-182 (Table 1), are both protective, working the first on protein induction of inflammasome activation, the second by alleviating inflammation and inducing polarization of macrophages [62,63]. A careful analysis of many studies concluded that MSCs and their EVs protect not only infarcts but also other types of heart diseases [64]. Subsequent studies carried out on both cord blood and adipose tissue MSC-EVs demonstrated that cardio-protective miRNAs, such as miR-22-3p (Table 1), are numerous, however they operate in parallel to miRNAs of different function. This could be a mechanism governing the multiple effects induced by EVs when administered to distinct types of cells [65].

Analysis was extended also to EVs involved in angiogenesis, which were found to contain many miRNAs. Two such miRNAs were found to induce an efficient stimulation of migration and invasion of endosomal cells, followed by growth of vessels [66]. Another miRNA, miR221 (Table 1), when administered to mature mice was found to affect the structure of vessels preventing in particular their atherosclerotic plaque formation [67]. Additional EV miRNAs were connected not to the processes mentioned so far, but to other processes. By the use of distinct miRNAs the EVs appear therefore to govern several processes of various function.

### 5.3. Kidney

This organ is affected by various treatments inducing severe cell lesion. Without therapy the affected kidney needs to be replaced by surgery. In various types of kidney diseases treatment with MSC-EVs from a few stromas induce significant protective effects sustained by various miRNAs. This has occurred in kidneys previously exposed to ischemia in which administration of EVs from umbilical cord in the course of reperfusion was found protective against apoptosis and inflammation, with ensuing reduction of tissue damages combined to regeneration. In this case the critical miRNA has been reported to be miR-93-5p (Table 1) [68,69].

During senescence MSCs are believed to play an important role in the prevention of kidney tissue fibrosis. An attempt was made in old mice to establish whether MSC-EVs induce a protective effect. The level of the vesicles and of their miRNAs, miR-294 and miR133b (Table 1), is low in the kidney of old mice. During treatment with MSC-EVs the two miRNAs induce an important protective effect on renal fibrosis of old rats [70]. In other prolonged kidney lesions the effects of the two miRNAs appear similar. However, in the kidney form of diabetes the level of vesicles and its critical miRNA, miR-26a-5p, is low. Upon treatment with adipose tissue MSC-EVs, miR-26a-5p increases its level (Table 1) and is transferred to mouse glomerular podocytes, with ameliorated state of the kidney [71]. In conclusion, the kidney operates under the control of MSC and its EVs, which play important roles on the state of the cells, both during short and long-term diseases.

### 5.4. Diseases and Other Lesions of the Brain

The long-term diseases of great importance to be considered are the Alzheimer’s disease (AD) and the other neurodegenerative diseases for which therapy is a severe problem [72]. NSCs, the neural cells of the MSC type, are generated and distributed in the brain stromas. Their EVs have been shown to play local roles analogous to those of MSCs outside the brain [73]. The changes induced in both AD and Parkinson’s mouse models, initially reported several years ago with involvement of at least one miRNA, miR-146 (Table 1), induce changes of various symptoms including astrocyte inflammation, synaptogenesis, reduced Aβ/α-synuclein deposition and cognitive impairment [74,75,76]. Progress of translational applications based on these effects is still in a relative infancy. However, its stem technologies hold considerable promise to combat neurodegenerative diseases [72].

In addition to neurodegenerative diseases, MSC-EVs protect against a number of other brain lesions. In the relapsing-remitting form of multiple sclerosis MSC-EVs (miR 467F and 466q, Table 1), have been shown to suppress proliferation of activated macrophages, contributing to the elimination of the episodes [77]. At present these studies are near to evolve from preclinical to clinical levels [78]. Post-stroke neurodegeneration is reduced by miRNAs even when administered after several days (miR-124, Table 1). Also post-ischemic immunosuppression is attenuated by MSC-EDs administered for a few days. All together the results provide evidence for future clinical studies in strokes of human patients [79,80]. Analogously, EV miRNAs from bone marrow, adipocytes and other stroma cells have been shown to protect spinal cord injuries [81], brain hemorrahagies [82], neuro-inflammation [83], deep circulatory arrest (miR-214, Table 1) [84], and other brain lesions.

### 5.5. COVID-19 Disease

COVID-19, a viral disease rapidly evolved into a pandemia. For over a year a main problem has been the lack of efficient drugs, a problem that now may move into solution [85,86]. In the meantime several vaccines have been analyzed and made usable. As long as vaccination is incomplete in the advanced countries and absent in the rest of the world, treatment with cargo molecules from MSC-EVs is still needed. Such treatments, inactive against the CAVID-19 virus SARS-CAV-2, are known to reduce the severe lesions of lung (miR-20a-5p, Table 1) and other organs induced during the disease [87,88].

The severity of the COVID-19 disease is mostly dependent on the patient responses. Over-activation of the immune system, developed in the attempt to kill the SARS-CAV-2 virus, can cause a “cytokine storm” which in turn can induce an acute respiratory distress syndrome (ARDS: [44,89]), a multi-organ damage, ultimately leading to death (miR-258, miR-266, Table 1). In COVID-19 patients, the immunomodulatory properties of MSCs ameliorate the cytokine storm by inhibiting or modulating the pathological events, especially those of severe cases [87,88,89]. Protection by MSCs is due to the release of their EVs, working via immunomodulatory effects, striking the COVID-19 balance by increased cell safety and tissue penetration [89]. The result by MSC-EVs could thus modulate the inflammatory responses of infected patients, promoting tissue-repair and regeneration of damaged organs [87,90,91].

**Table 1 biomedicines-09-00667-t001:** MSC-EVs affecting diseases and processes in various organs by the action of specific cargo mRNAs.

Organs (Years of the First Report)	Diseases & Processes	Examples of Active miRNAs within the MSC-EVs Involved
Bones (2011)	osteoblastogenesis	miR-133-3p [59]
	rheumatic diseases	miR-483-5p [60]
	bone regeneration	miR-378 [61]
Heart & Vessels (2012)	infarction cardio-protection atheroslerosis	miR-125b [62]
		miR-182 [63]
		miR-22-3p [65]
		miR-221 [67]
Kidney (2011)	ischemia	miR-93-5p [69]
	senescence	miR-133b [70]
	diabetes	miR-26a-5p [71]
Brain (2013)	neurodegeneration multiple sclerosis stroke circulation arrest	miR-146 [74]
		miR-467f [83]
		miR-466q [83]
		miR-124 [79]
		miR-214 [84]
Lung (COVID-19) (2020)	cytokine storm/ARDS	miR-258 [87]
	pneumonia	miR-20a-5p *
	cell death	miR-266 [92]

The data shown about miRNAs were from the quoted articles, except for * = Li C.X. et al. Mediators Inflamm. 2021; 2021:6635925.

Among the molecules of the MSC-EV cargo inducing alleviations of inflammatory responses there are several miRNAs. Some of them, known to exacerbate action of cytokines and chemokines, are down-regulated by EVs, while others, that modulate the above processes, tend to prevent tissue damage. The heterogeneity of EV cargo molecules is therefore relevant for the survival of CAVID-19 patients [91,92]. In other studies, SARS-CaV-2 is affected in the hippocampus by brain vesicles, the RSC-EVs. Such EVs have a cell-free action in which viruses are degraded by an adaptive function [73]. In conclusion, MSC-EVs have been recognized as the best among the EV attempts of COVID-19 protection, however its practical employment should be delimited [93]. In the next Section of this review we it will mentioned in terms of therapy [93,94].

## 6. Therapy

The high interest for MSC-EVs, documented in the presentation of diseases reported in the previous Section, depends on their therapeutic potential [95], reported in many of the articles quoted in this review. For many years therapy had been presented as an advantage attributed to the parental cells. In patients, however, such advantage had been questioned. In addition, it was accompanied by risks including toxicity and immunogenicity challenges, immune rejections and cancers, as already mentioned [20,21,22,23]. At present, therefore, therapies by MSCs are employed only in a few diseases for which advantageous alternatives are not available (for examples see [18,19,20,88]). In most cases, however intense studies of the last several years have lead to the demonstration that MSC-EVs are the advantageous alternatives that were needed. On the one hand, in fact, they recapitulate all the properties analogous, and some times even better than those of their parental cells; on the other hand, the therapies by MSC-EVs are employed without association to negative risks related to the structure and function of their parental cells (Figure 1) [24,54,55,95,96].

The properties of MSCs recapitulated by EVs include the regulations of immune responses. Paracrine recruitment of MSC-EVs in the proximity of needed cells may result in boosting immune responses, associated in many cases to protective roles in infectious diseases [23,24,45,46,47,60,64,82,83]. Such properties are due to interaction of the vesicles with immune cells, such as T lymphocytes, B lymphocytes, natural killer cells, dendritic cells, and macrophages. Among these cells the ones that play an essential role in innate immunity, adaptive immunity, and homeostasis, are target macrophages. Recent studies have demonstrated that MSC-EVs reduce their M1 reactive polarization and/or promote their immunosuppressive M2 polarization in a variety of settings. The results of such changes affect a number of cell-free therapies addressed to several diseases including cardiovascular, pulmonary, digestive, renal, immunological, cancer and central nervous system diseases [48,97,98,99].

So far we have considered the therapy produced by the general capacity of MSC-EVs. In addition, a refined mechanism of therapy has been developed for cells and vesicles decades ago according to engineering approaches. Such approaches are based on a good manufacturing practice (GMP), relevant in the selection of materials, the manufacturing and the quality assays employed [100,101,102]. By such approaches the cells/vesicles are engineered by insertion of changes both at the surface and within the lumen, assuring preloading of drugs followed by binding and fusion to appropriate target cells, with discharge of drugs within their cytoplasm. In spite of the efforts made during the last decades, the use of these artificial approaches in diseases has been limited. In contrast, MSCs and, even better, the MSC-EVs have resulted much more successful due to the combination of engineered and natural approaches, efficient especially in inflammation treatments. For this the induced engineering changes need to be moderate, while the EV properties need to be functional, competent for cell fusion and cargo diffusion, typical processes by which key molecules flow from EVs to target cells [103,104]. The therapy combination summarized so far is effective also for the lesions induced by COVID-19 in the lung and other organs. In these cases MSC-EV operate in nano-platforms for therapeutics and drug delivery to combat COVID-19.

## 7. Clinical Practice

While research with MSC-EVs has exploded during the last few years, its development towards clinical practice [20,28,54] is still early. At present we intend to consider the problem from two points of view: technical and operative properties of MSC-EVs considered during the last few years for entrance in clinical practice; and the state of science for MSC-EV therapy in various diseases.

In the first we summarize the requirements for manufacturing, safety and efficacy of clinical practice by testing the sources of cell types and their target diseases. At present the biotechnological and pharmaceutical companies are considering with interest the opportunity to invest in this type of initiatives. For such interest it will become important to identify animal models appropriate for studies and operations. Preclinical work, already considered in previously mentioned studies [30,31,32,80], is relevant for the choice of approaches, the toxicities to be considered, and the pharmacological properties to be obtained. Manufacturing of the molecules will be started based on the diseases considered, keeping in mind also the number of their potential patients. The size of EV planning will thus be established. Upon their characterization EVs will be purified and analyzed, first for their pharmacological consideration, and then for their production [105,106].

The form often employed to strengthen the early development of clinical practice is regenerative medicine, based on the EV-dependent regeneration of affected tissues such as the systems of brain, blood vessels and others [28,50,64,71,72,107,108,109,110]. Critical for these processes, operative by cell-free strategy, is the successful translation of preclinical studies into clinical platforms [21,32,45]. Aspects to carefully evaluate the findings concern qualification, characterization and production of the methods employed; pharmacokinetics, targeting and transfer of drugs to appropriate sites; assessment of safety profiles [108]. Regenerative medicine of various diseases often takes place by encapsulated drugs administered by various procedures including direct, intravenous and intraperitoneal injection as well as oral and nasal delivery [96,102,104,105,106,107,108,109].

## 8. Conclusions

The task of this review has been a comprehensive illustration of the present knowledge about MSC-EVs, the extracellular vesicles known to express many important properties of their parental cells, including their therapeutic activity. Based on the combination of such properties to EV peculiarities, including their lack of self-replication, differentiation, and risk factors, MSC-EVs are now widely recognized for the advantageous replacement of their parental cells in the therapy of numerous diseases. Such unique role explains the extraordinary success and the intense studies dedicated to these vesicles.

Many properties of MSC-EVs, including their therapy for diseases, have been already illustrated in detail in the present review. In contrast, we have not mentioned a number of other properties not unique to MSC-EVs but common to other vesicles, secreted by parental cells different from MSCs. A recent study has emphasized many such properties, necessary for all types of EVs to be distinguished from the other organelles of the cells [3]. Here we will mention three of such properties: the long survival of EVs after their secretion; their withstanding of harsh environmental conditions; their unique ability to cross the blood-brain- barrier, and thus to move in and out the brain [3]. In our opinion these, and possibly other properties we have no space to emphasize, establish general conditions appropriate to the specific activities of MSC-EVs.

Another important aspect, mentioned repeatedly however without conclusion, concerns the mechanisms of EV action, i.e., the processes that follow their binding to target cells, including their following fusion with discharge of their luminal cargos into the cytoplasm [2]. Many key agents of vesicle function are concentrated in their lumen. In addition to proteins, lipids, coding RNAs and other agents, the vesicles contain miRNAs. These non-coding nucleotides are often present in considerable number, larger than those believed a few years ago. Many if not all such miRNAs are needed for the activation of signaling cascades leading to the final responses of target cells. In some cases various steps of whole cascades have been discovered. For example, increased miR-410 induces inhibition of HDAC1, a modulator of negative gene transcription, and activation of an axis containing EGR2, a transcription factor, and Bcl2, a protein that supports cell survival. By impeding neuronal apoptosis, the axis appears to prevent a brain lesion of newborns, the hypoxia-ischemia brain damage [111]. Identification of the three such steps looks valid, however the cascade appears long and some possible likely steps remain unknown.

The previous paragraphs of this Conclusion have illustrated two fields in which MSC-EV investigation appears oriented. Additional fields could be related to critical Sections of our presentation: immunity, diseases, therapy. However, the main task will be the expansion and strengthening of clinical practice, i.e., a successful conversion of knowledge about MSC-EVs into various aspects of modern medicine. Development of knowledge is already excellent, and important improvements are expected for the next future. The potential development of clinical practice appears therefore highly promising.

## Figures and Tables

**Figure 1 biomedicines-09-00667-f001:**
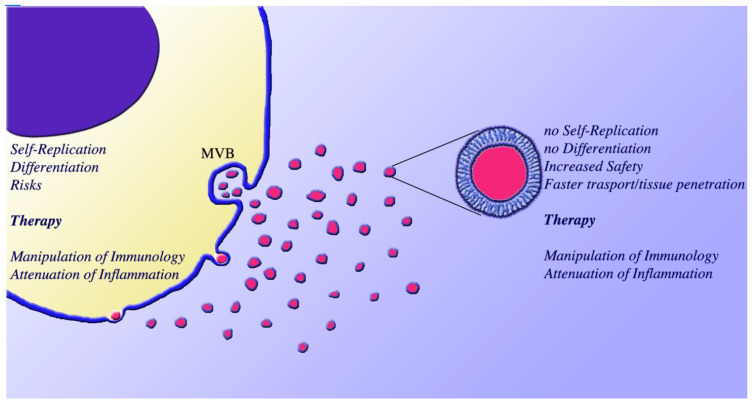
Comparison of MSC cells and their EVs. To the left an MSC cell illustrates its secretion of EVs. The small exosomes, generated and accumulated within endosomal vacuoles, the multi vesicular bodies (MVBs), are released upon exocytosis of the latter. The ectosomes, assembled in direct contact to the cytosolic surface of specialized rafts in the plasma membrane, undergo pinching off and then shedding to the extracellular space. A list of the MSC properties is shown to the left, over the cell cytoplasm (yellow). The first two, self-replication and differentiation, are typical of many types of cells active as MSCs. These two activities open the cells to many risks including cancers and immune rejections. The main property of MSCs is natural therapy against many diseases based, among others, on the properties of immunology manipulation and attenuation of inflammation. However, due to the dangers of accompanying risks, the therapeutic approach of MSCs in humans is limited to a few types of cells and diseases. The MSC-EVs, shown to the right, are the model of therapy alternative to MSCs. MSC-EVs recapitulate many properties analogous, and some time even better than those of the parental cells, including those in immunology and inflammation. Due to their different strategy, MSC-EVs do not express many properties related to the structure and function of their parental cells. In fact, they show no replication, no differentiation and no few other properties. Increased safety in the extracellular space, faster transport and good tissue penetration are advantages of MSC-EVs, especially when administered in vivo. Finally, the MSC-EVs are more appropriate for GOP engineering (see Section 5—Diseases: Mechanisms of Protection by MSC-EV). The ensuing, technical therapy can be combined to the natural forms expressed by the vesicles. The properties of MSC-EVs, illustrated here, together with others reported and discussed in this review, explain the extraordinary success of the intense studies dedicated to these vesicles, especially those about therapy.

## Data Availability

The data of this review will be available to all scientists of scientific interest.

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
