# Peer review of "Extracellular Vesicles of Mesenchymal Stem Cells: Therapeutic Properties Discovered with Extraordinary Success"

_biomedicines, 2021, doi:10.3390/biomedicines9060667_

Round 1

Reviewer 1 Report

In this manuscript, which is a review article, authors discuss the therapeutic potential of mesenchymal stem cell-derived extracellular vesicles (MSC-EVs).

The review is interesting and has a scope. While authors cover a topic of interest, there are certain points which need to be revised.

My comments and suggestions are appended below.

(1). In the abstract include a mechanistic insight

(2). In the main body of the manuscript, discuss the functions of MSCs, in terms of stem cell biology (e.g, MSC isolation, their markers, contribution in self-renewal, and repair).

- Then mention the risks associated with MSC, teratoma formation, risk of graft rejection, immune reactions, and so…).

- Then how EVs are advantageous over these risks.

(3). While authors indicate the recapitulating roles of MSC-EVs in stem cell biology, and organ repair, I recommend authors to cite the following articles (PMID: 26649044 and PMID: 29123544).

(4). Since main purpose of the current review is to discuss role of MSC-EVs in different diseases, organ protection and therapeutic potential of MSC-EVs, it is imperative to include a schematic showing the mechanistic insights how MSC-EVs act on organ protection, and how they can be implicated as therapeutic modalities. Show a schematic illustration.

(5). Each review should provide a balanced view of the subject, therefore, I suggest authors to include other side of EVs also. i.e. possible risks, regulatory affairs, manufacturing and other considerations. Please see PMID: 28587212 and PMID: 32425691.

Other minor:

There are minor typo errors e.g, extra spacing or no spacing, and some extra characters in the headings. This needs a proofreading. 

Author Response

Reviewer 1

Thank you for the interest you have shown in our review. We have been pleased to read your numerous comments we have received, many of which have been useful to improve our work. Specifically our answers will be listed according to the list of your questions/criticisms. After such specific report you will find 3 changes we propose. These changes have nothing to do with your report but we intend you to know. They will be sent to the Office of the Journal because, for the specified reasons, we wish them to be introduced in the final version of the paper

Answer to your comments

  1. Include In the abstract a mechanistic insight.   We have included a mechanistic insight in the Abstract, as recommended.
  2. Discuss the functions of MSCs, in terms of stem cell biology: risks and EV advantages. We have extensively rewritten the Introduction as well as the Sections 2 and 3 of our review, introducing details about “the functions of MSCs in terms of cell biology”, as requested. In the original version this information was short and also inaccurate. In our opinion, therefore, the recommended changes we have included are appropriate. On the other hand we emphasize that, as specified in the title and also at the end of the Introduction, our work deals primarily with the EVs, focusing in particular on the developments “recognized during the last few years”. Thus, presentation of MSCs is important for the introduction of the review. It cannot predominate because their interesting information is not recent, and because the focus of our review is about EVs. In particular, the key roles of EVs deal with their diseases and therapy, with marked decrease of their risks, as it is now clearly specified in the review.  
  3. Roles of MSC-EVs in stem cell biology, and organ repair. Two articles recommended.   We have strengthened the presentation of tissue regenerations, including the two references recommended, now labeled as references 20 and 26.
  4. Include a schematic(figure) showing the mechanistic insights how MSC-EVs act on organ protection, and how they can be implicated as therapeutic modalities.  The reviewer emphasized, as the main purpose of our review, “the discussion of MSC-EVs in different diseases”. Thus she/he recommends “as an imperative” a new Figure about “mechanistic insights on organ protection and therapeutic modalities” of the various diseases. In theory we agree with the Reviewer. In fact, even before receiving the review, we have considered the possibility to include in the paper a Figure about many mechanisms of EV function. In our opinion, however, such Figure is almost not feasible. As specified in a very recent paper by one of the groups recommended by the Reviewer (Gimona et al., Cytotherapy 2021; 23: 373-380) “assays should ideally reflect the mechanism of action (MoA) of MSC-EVs. This however is challenging because, for the reported efficacy of MSC-EV preparations against multiple diseases of diverse underlying pathology, the MoA is likely to be complex and different for each disease, thus difficult to be fully elucidated”. In our opinion, the mechanism feasible at present includes miRNAs and the ensuing cascades of very important active proteins: factors, enzymes and others. We have considered this mechanisms by including miRNAs in Table 1 and, at the end of the review, introducing an example of cascade triggered by the activation of a miRNA. At present we do not think such issue can be extended any more.
  5. Include other side of EVs also. i.e. possible risks, regulatory affairs, manufacturing and other considerations. Two articles recommended.

We have included some balance between the sides of EV action. Both papers recommended in this area by the Reviewer are now included in the Reference list (references 93 and 106). Risks are now presented in much more detail than in the first version. Regulatory and manufacturing are briefly presented in a new paragraph we have included in the Clinical Medicine Section 7, at present twice as large as in the original version.

   Minor point: minor type errors corrected.

We hope that the new version of our review, largely revised according to criticisms of the Reviewer, will be appreciated. Thanks for the detailed and accurate revision made.

Following are the 3 changes that have nothing to do with your review but we intend you to know. We wish these changes to be made in the final version of the paper for the reasons specified here.

  1. In the text I have received the Legend of Fig. 1 was missing. I have introduced such Legend and removed the duplicated portion of the text found in its space.
  2. The initial version of the review included a subsection 5E dealing with cancers. We have decided that the presentation was not appropriate and therefore we have removed it, specifying that it will be presented elsewhere.
  3. 1 and Table 1 are included too early in the text. We recommend they to be moved, Fig. 1 from page 2 at least to page 3; Table 1 from page 5 to page 7.

Reviewer 2 Report

In the review article entitled “Extracellular Vesicles of Mesenchymal Stem Cells:Therapeutic Properties Discovered with Extraordinary Success” the authors aimed to present current therapeutical perspectives of MSC-derived extracellular vesicles in the light of the current literature.

Based on the title of the article I would expect that in the manuscript we get an overview about the current mesenchymal stem-cell (MSC) and MSC-derived EV based preclinical and clinical data. However, preclinical and clinical data are overmixed, and most importantly the terms mesenchymal stromal cells and mesenchymal stem cells are used interchangeably, however, they are not synonyms!

The EV-associated cargo is mainly addressed at miRNA level only. The figures should be better designed, e.g. they do not reflect the difference between small- and medium-sized EVs. Overall I feel the article should focus on specific questions, rather than address

I would have the following major points:

  1. I would suggest the use of EV subtypes according to MISEV2018 guidelines: instead of exosomes/ Exos please use small EVs (sEV, enriched in exosomes) and instead of microvesicles I would suggest the use of intermedier-sized/ medium-sized EVs (iEV/mEV enriched in microvesicles). According to my suggestion please update in the manuscript the nomenclature of EV subtypes and cite MISEV2018 paper. Also please use current data, depicting the biogenesis of EVs, in the current form it is obsolete.

Clotilde Théry et al. (2018) Minimal information for studies of extracellular vesicles 2018 (MISEV2018): a position statement of the International Society for Extracellular Vesicles and update of the MISEV2014 guidelines, Journal of Extracellular Vesicles, 7:1, DOI: 10.1080/20013078.2018.1535750

  1. Since the review addresses the mesenchymal stem cell derived-EVs, it would be advisable to cite papers and detail the effects of mesenchymal steml cell derived-EVs only, and do not include data about the mesenchymal stromal cells, unless this would be detailed explicitly. It would be useful to follow the recommendation of the latest position paper released by the International Cell and Gene Therapy.

Viswanathan, S., Shi, Y., Galipeau, J., Krampera, M., Leblanc, K., Martin, I., Nolta, J., Phinney, D. G., & Sensebe, L. (2019). Mesenchymal stem versus stromal cells: International Society for Cell & Gene Therapy (ISCT®) Mesenchymal Stromal Cell committee position statement on nomenclature. Cytotherapy, 21(10), 1019–1024. https://doi.org/10.1016/j.jcyt.2019.08.002

  1. In the introduction part, addressing EV-isolation/ enrichment methods should be improved, technical details, which enhance and are scalable for therapeutic approach should be discussed, presented. Also “Attempts to induce separate isolation of the two types of vesicles have often been unsuccessful.” is not true, however it is quite challenging to obtain pure EV fractions. This should be rephrased accordingly.

  1. Figure 1: Please use smaller EVs for MVB exocytosis, and larger for plasma membrane-derived EVs. In Figure 1 description - I recommend to rephrase it, since EVs can elicit endocrine, paracrine and autocrine effects.

I have a few other minor remarks:

- Page 4 LINE 3: use “in vitro” instead of “in culture”

- I suggest the use of “preclinical”  instead of “pre-clinical” in the manuscript e.g. page 7

- Nomenclature used for miRNAs should be the same in the manuscript e.g. miR-221 and not miR221

Author Response

Reviewer 2

We thank for the detailed evaluation by the Reviewer that has helped us in the improvement of our review. We are convinced that the revised version already completed is greatly improved with respect to the first version. After such specific report you will find 3 changes that will be sent to the Office of the Journal. These changes have nothing to do with your report but we intend you to know. For the specified reasons, we wish them to be introduced in the final version of the paper.

Concerning the initial comments presented by the Reviewer before the major points we specify that

  1. Mixed preclinical and clinical data . Before completing the review we had expected a larger contribution by clinical data. However, in the detailed evaluation of the papers appeared during the last three years we only found the definition of “early, specified in limited fashion”. This is reported accurately in initial Sections and in the specific Section 7, which has been enlarged. In our opinion, therefore, the problem is not ”a mixture of preclinical and clinical data”, it is a reasonable presentation.
  2. Interchangeable presentation of stem and stromal cells. We have not mixed the definitions of stem and stromal cells. Our definition is only stem cells, stromal appears only in the Reference List in the title of papers by others.
  3. The EV cargo. In our presentation the molecules contained within the lumena od EVs are specified: proteins, lipids and RNAs. The problem is that the molecules mostly reported as controllers of EV function are the miRNAs. Elimination of some such miRNAs has been shown to block the specific effects. Thus miRNAs are believed to participate in the control, working as controllers of specific cascades of factors, enzymes and other proteins. No similar control effects have been reported for the other cargo components.

Major points

  1. Nomenclature of EVs The definition of exosomes and ectosomes as sEVs and lEVs is now reported in the Introduction of the review. In the MSC-EV literature of the last 3 years, however, a significant fraction of the vesicles employed are mixtures of exosomes and ectosomes. As specified also in the Introduction our review deals with these vesicles named EVs. What is wrong with this choice? Our work has been done accordingly. To report about the nomenclature we have not use the paper by Thiery but one by Van Niel, D’Angelo and Raposo published in Rev. Mol. Cell Biol. 2018.

  1. Mesenchymal stem cell derived-EVs.   We agree that our present review deals with mesenchymal stem cell-EVs. As already mentioned in B, our text does not includes mention to mesenchymal stromal cells.
  2. EV-isolation/ enrichment methods.  Here the Reviewer is completely right, the Introduction has been largely rewritten specifying as recommended.
  3. Figure 1 . Again, the Reviewer is right, however the problem was not in our presentation, rather on the conversion of the review from our version to the one of the Journal , in which the Figure Legend was not included. In the present revised version the Legend has been inserted in the correct position. Analysis of a Figure without legend was certainly very annoying for Reviewers, we agree.

Minor remarks

In vitro has been changed into in culture, as requested

Preclinical is used as requested

We apologize for the variable presentation of miRNAs, now all of them are presented as as miR-number, as recommended.

We hope that the new version of our review, largely revised according to the criticisms of the Reviewer, will be appreciated. Thanks for the detailed and accurate revision made.

Following are the 3 changes that have nothing to do with your review but we intend you to know. We wish these changes to be made in the final version of the paper for the reasons specified here.

1    In the text I have received the Legend of Fig. 1 was missing. I have introduced such Legend and removed the duplicated portion of the text found in its space.

  1. The initial version of the review included a subsection 5E dealing with cancers. We have decided that the presentation was not appropriate and therefore we have removed it, specifying that it will be presented elsewhere.
  2. Fig. 1 and Table 1 are included too early in the text. We recommend they to be moved, Fig. 1 from page 2 at least to page 3; Table 1 from page 5 to page 7.

Round 2

Reviewer 1 Report

The authors have carefully revised their manuscript. The revised version of the review is much improved and suitable for publication. 

I endorse the production of this work.